# Biodegradation of Poly(Lactic Acid) Biocomposites under Controlled Composting Conditions and Freshwater Biotope

**DOI:** 10.3390/polym13040594

**Published:** 2021-02-16

**Authors:** Pavel Brdlík, Martin Borůvka, Luboš Běhálek, Petr Lenfeld

**Affiliations:** Faculty of Mechanical Engineering, Technical University of Liberec, Studentska 1402/2, 46117 Liberec, Czech Republic; martin.boruvka@tul.cz (M.B.); lubos.behalek@tul.cz (L.B.); petr.lenfeld@tul.cz (P.L.)

**Keywords:** biocomposites, poly(lactic acid), plasticiser, CaCO_3_, nano-cellulose, aerobic biodegradation, freshwater biotope

## Abstract

The influence of additives such as natural-based plasticiser acetyl tributyl citrate (ATBC), CaCO_3_ and lignin-coated cellulose nanocrystals (L-CNC) on the biodegradation of polylactic acid (PLA) biocomposites was studied by monitoring microbial metabolic activity through respirometry. Ternary biocomposites and control samples were processed by a twin-screw extruder equipped with a flat film die. Commonly available compost was used for the determination of the ultimate aerobic biodegradability of PLA biocomposites under controlled composting conditions (ISO 14855-1). In addition, the hydro-degradability of prepared films in a freshwater biotope was analysed. To determine the efficiency of hydro-degradation, qualitative analyses (SEM, DSC, TGA and FTIR) were conducted. The results showed obvious differences in the degradation rate of PLA biocomposites. The application of ATBC at 10 wt.% loading increased the biodegradation rate of PLA. The addition of 10 wt.% of CaCO_3_ into the plasticised PLA matrix ensured an even higher degradation rate at aerobic thermophilic composting conditions. In such samples (PLA/ATBC/CaCO_3_), 94% biodegradation in 60 days was observed. In contrast, neat PLA exposed to the same conditions achieved only 16% biodegradation. Slightly inhibited microorganism activity was also observed for ternary PLA biocomposites containing L-CNC (1 wt.% loading). The results of qualitative analyses of degradation in a freshwater biotope confirmed increased biodegradation potential of ternary biocomposites containing both CaCO_3_ and ATBC. Significant differences in the chemical and structural compositions of PLA biocomposites were found in the evaluated period of three months.

## 1. Introduction

Environmental pollution caused by petroleum-based plastic products is one of the anthropogenic impacts that our planet and its ecosystems face today. Therefore, focusing on the development of biodegradable materials from renewable sources is tremendously important. There are currently several commonly available biopolymers on the market [1,2]. Polylactic acid (PLA), due to its relatively good barrier properties to aromas and high mechanical strength and modulus, is one of the most used biopolymers today [3,4]. Unfortunately, there are also some limits that restrict its applicability, particularly its low thermal resistance and high brittleness. There are several modifications that can be used to treat these limits [5]. In the packing industry, low flexibility and resistance to fracture are very often optimised by the addition of plasticisers [6]. Phthalic acid esters have become the most-commonly used class of plasticisers in the 21st century [7]. However, a problem regarding the migration of phthalates and its negative impact on human health has been presented in several works [6,7,8]. Consequently, there is increasing interest in the development of new eco-friendly natural-based plasticisers. Various types of plasticisers have already been tested with PLA, such as poly(ethylene glycol) (PEG), citrate esters [9,10,11], oligomeric lactic acid and triacetin [12]. Other limits, such as the low thermal resistance and other mechanical and barrier properties, can be improved through an increase in crystallisation kinetics [13,14]. One very effective way to do so is increasing nucleation density by adding a heterogeneous nucleation agent [15]. The synergistic effect of heterogeneous nucleation and increased chain mobility due to the addition of both nucleation agents and plasticisers can positively affect the crystallinity degree of PLA [16].

The degradation of biopolymers is generally assumed to be excellent. However, the degradation rate varies among these materials [17]. The scission of the main and/or side chains of the macromolecules can be initiated by a process such as photolysis, oxidation, radiolysis, thermal activation, hydrolysis or biological activity [18]. For polyesters, the hydrolysis and microbial/enzymatic activity are the most important mechanisms [18]. Therefore, medium characteristics (e.g., temperature, pH, the presence and flow of oxygen) are tremendously important in the biodegradation process. Temperature, in particular, is a crucial factor. When PLA is exposed to higher temperatures than the glass transition temperature, the flexibility of the polymer chain is increased. Consequently, PLA easily facilitates the hydrolysis reaction and the attachment of microbes/enzymes [19]. Itävaara et al. [20] reported on the influence of temperature on aerobic and anaerobic biodegradation processes. Their results showed that higher levels of biodegradation were achieved at higher temperatures. Moreover, Kolstad et al. [21] and Karlssion et al. [22] found significant differences in the biodegradation of PLA exposed to mesophilic and thermophilic conditions. Nevertheless, temperatures higher than thermophilic conditions (45–65 °C) are not desirable as they result in the death of microorganisms. Alkaline conditions during composting may also enhance hydrolysis and, thus, biodegradation [20]. The pH of the medium is another characteristic worthy of consideration. For most microorganisms and enzymes, a pH-neutral medium is optimal. During the biodegradation process, the polymer itself can influence the pH of the medium. Kale et al. [23] found that the PLA during composting caused production of metabolites, resulting in decreased pH and, thus, lower levels of microorganism activity. Chemical composition, stereoregular conformation of the lactide group, copolymerisation, blending and properties such as molecular mass, density and crystallinity degree are other major aspects in the PLA biodegradation process. Cadar et al. [24] reported some correlation in the composting and level of biodegradation of PLA-based copolymers. The biodegradation of copolymers containing higher amounts of lactic acid was found to be faster than the biodegradation of copolymers containing smaller amounts. Similar results were reported by Iovino et al. [25] and Soni et al. [26]. Furthermore, Kolstad et al. [21] reported very poor biodegradability with semicrystalline PLA when compared to amorphous PLA. This is because water molecules easily diffuse into amorphous regions, and these regions are also easily assimilated by microorganisms [27]. In the research of Kallel et al. [28], a positive effect on biodegradation behaviour was detected following the blending of PLA with PEG. Furthermore, Samadi et al. [17] reported increased biodegradation following the blending of highly degradable PEG with PLA.

Regarding the previous summary, plenty of studies concerned with the PLA biodegradation process have been published. However, the influence of additives such as plasticisers and heterogeneous nucleation agents has still not been sufficiently explained. Therefore, the current work is dedicated to evaluating the influence of bio-based plasticiser acetyl tributyl citrate (ATBC), spray-dried lignin-coated cellulose nanocrystals (L-CNC) and calcium carbonate (CaCO_3_) on the biodegradation behaviour of PLA. As this behaviour is considerably influenced by the medium’s characteristics, two different biodegradation tests were performed. The first was concerned with the determination of the ultimate aerobic biodegradability under controlled composting conditions at 58 °C (ISO 14855-1), while the second was focused on the analysis of biodegradability in the freshwater biotope (25 °C).

## 2. Materials and Methods

The commercial PLA under the trade name of Ingeo 3001D was supplied by Nature Works (Minnetonka, MN, USA). It is a material that contains 95 wt.% of L-lactide and has an approximate molecular mass of 155,000 g/mol, glass transition temperature of 60 °C and melting point of 170 °C. Furthermore, ATBC under the trade name of Citroflex A-4 (Vertellus Holding LLC, Indianapolis, IN, USA) was used in the form of an oily liquid as a plasticiser. Spray-dried L-CNC (BioPlus-L^TM^ Crystals, Orlando, FA, USA) with an average particle size of 4–5 nm in width and 500 nm in length were purchased from American Process Inc. (Atlanta, GA, USA) and used as a nucleation agent. Precipitated calcium carbonate (CaCO_3_) (Honeywell Fluka, Seelze, Germany) with an average particle size lower than 1 µm was also used as a nucleation agent. 

### 2.1. Preparation of PLA Films 

The PLA pellets, as well as the L-CNC and CaCO_3_ nucleation additives, were dried for 24 h at 50 °C in vacuum oven VD53 (Binder GmbH, Tuttlingen, Germany) before being processed to remove eventual moisture. Concerning the maximal dispersion and distribution, the ternary biocomposites (PLA/ATBC/L-CNC, PLA/ATBC/CaCO_3_) and plasticised PLA (PLA/ATBC) were compounded prior to film processing. The laboratory micro-compounder MC 15 HT (Xplore, Sittard, Netherlands) with conical screws, a speed of 100 rpm and a constant temperature profile of 180 °C was used as the primary processing device. The dispersion and distribution were controlled by the level of torque. Before film processing, the biocomposite compounds (pellets) were again placed in a vacuum oven and dried at 50 °C for 24 h. The dried pellets were further extruded on the twin-screw extruder MC 15 HT with flat film die (0.2 mm gap size) at a melt temperature of 180 °C and 80 rpm screw speed. The extruded films were drawn by cooled rolls. The individual compositions of the prepared biocomposite films are listed in Table 1. 

### 2.2. Determination of Ultimate Aerobic Biodegradability under Controlled Composting

Respirometric tests were conducted using a method adapted from the ISO 14855-1 standard to evaluate the biodegradation kinetics of PLA biocomposite films in composting conditions. This method is based on the measurement of the amount of carbon dioxide evolved during microbial degradation. For this purpose, spirometer ECHO (ECHO d.o.o., Slovenske Konjce, Slovenia) with automatic leak detection, precise mass flow controllers and automatic humidification was used. Biodegradation tests were performed in 2.8 L cylindrical hermetic vessels containing 100 mL of demineralised water (for humidification of inoculum), 150 g of compost and 10 g of PLA tested films that were trimmed to individual pieces with sizes of about 1 cm × 1 cm. The commonly available compost from the company AGRO CS (Říkov, Czech Republic) was used in this study. A pH of 6.3 was measured by the Voltcraft PH-100ATC pH meter (VOLCRAFT, Wollerau, Switzerland), while each vessel was determined to have 26.6 g content of volatile solids (550°/5 h) in the oven CLASIC 3014 (CLASIC CZ, Řevnice, Czech Republic). In accordance with the ISO 14855-1 standard, pebbles and foreign objects bigger than 2 mm were removed from the compost. Moreover, 50% humidity water content was adjusted by the halogen moisture analyser Mettler Toledo™ HX204 (Mettler Toledo, Columbus, OH, USA). The blank control was composed of only compost. Before use, the correct production of carbon dioxide in a blank control vessel was controlled. The inoculum in blank control vessel shall produce between 50 mg and 150 mg of CO_2_ per gram of volatile solids over the first 10 days (ISO 14855-1). The evaluated value for applied compost was 112 mg of CO_2_ per gram of volatile solids. Therefore, can be stated that there was significant initial microbial activity. Furthermore, microcrystalline cellulose with particle size of less than 20 µm (Sigma-Aldrich, Saint-Quentin-Falavier, France) was used as positive reference material to control the correct microbiological activity of the compost. Each biodegradation study was performed in duplicity at a constant temperature of 58 °C. The respirometer was shielded from the light. Every week, glass vessels were opened and the inoculum was stirred to ensure an even distribution of moisture. The percentage of biodegradation was determined from the cumulative amount of released carbon dioxide in accordance with the following equation: (1)$$D_{t}=\frac{{\left({\mathrm{CO}}_{2}\right)}_{T}-{\left({\mathrm{CO}}_{2}\right)}_{B}}{T_{{\mathrm{hCO}}_{2}}}\cdot 100$$
where (CO_2_)_T_ is the cumulative amount of carbon dioxide evolved in the composting vessel containing the test material, (CO_2_)_B_ is the mean cumulative amount of carbon dioxide evolved in the blank vessels and (T_hCO2_) is the theoretical amount of carbon dioxide that can be produced by the test material (all in g/vessel).

The theoretical amount of carbon dioxide can be determined via the following equation: (2)$$T_{{\mathrm{hCO}}_{2}}=M_{\mathrm{TOT}}{*C}_{\mathrm{TOT}}\cdot \frac{44}{12}$$
where M_TOT_ is the total dry solids in the test material introduced into the composting vessel at the start of the test (in g), C_TOT_ is the proportion of total organic carbon in the total dry solids in the test material (in g/g) and 44 and 12 are the molecular mass of carbon dioxide and the atomic mass of carbon, respectively. The individual proportions of total organic carbon of PLA biocomposite components are listed in Table 2. 

### 2.3. Analysis of Biodegradability in Freshwater Biotype

Biodegradation of PLA films was also studied in the freshwater biotope. Samples of the same size (5 mm × 15 mm) were placed in an aquarium containing a steady biotope (pH 8.1, temperature 25 °C) of aquatic plants, algae, fish, snails, etc. for 3 months. To determine the efficiency of biodegradation, qualitative analyses—including Fourier transform infrared spectroscopy (FT-IR), differential scanning calorimetry (DSC), thermogravimetric analysis (TGA) and the evaluation of surface images taken by scanning electron microscopy (SEM)—were performed. Due to the fast defragmentation of ternary biocomposites and plasticized PLA in the initial trial test, the calculation of biodegradation degree through a weight reduction was not considered. Another reason was negative influence of conditioning on microbial activity. Consequently, the samples were taken from biotope only for trimming of specimens for qualitative analyses. Trimmed specimens were conditioned (25°/96 h) before characterisation. The results from the introduced analyses were evaluated from 3 measurements. As the data set was small, only average values have been provided; thus, standard deviation has not been specified.

#### 2.3.1. Fourier Transform Infrared Spectroscopy (FT-IR)

The chemical changes of the PLA films were analysed using an infrared spectrometer Nicolet iS10 (Thermo Fisher Scientific, Waltham, MA, USA) in Attenuated Total Reflectance (ATR) mode and equipped with a diamond crystal. The FTIR-ATR spectra were recorded in the range of 400–4000 cm^−1^ by averaging 64 scans and using a resolution of 2 cm^−1^. 

#### 2.3.2. Differential Scanning Calorimetry (DSC)

Thermal properties were determined in a calorimeter DSC 1/ 700 (Mettler Toledo, Greifensee, Switzerland) under a constant nitrogen flow of 50 mL/min. Samples of approximately 5 mg were prepared from the cross-section of the PLA films and placed in an aluminium pan before being sealed and placed in the DSC chamber. The samples were heated from 0 °C to 200 °C with a heating rate of 10 °C/min. Further isothermal conditions were kept for 180 s to remove previous thermal history and then cooled again (10 °C/min). Primary glass transition temperature (T_g_), cold crystallisation temperatures and enthalpies (T_cc_, ∆H_cc_) and melting temperatures and enthalpies (T_p,m_, ∆H_m_) were evaluated. The degree of crystallinity (X_C_) was determined through the following equation, where $\Delta H_{m}^{O}$ is the melting enthalpy of 100% crystalline PLA (93.1 J/g) [2,29] and w_m_ is the mass fraction of PLA in the composites.
(3)$$X_{C}=\frac{\Delta H_{m}-\Delta H_{\mathrm{pc}}-\Delta H_{\mathrm{cc}}}{\Delta H_{m}^{O}\cdot w_{m}}\cdot 100$$

#### 2.3.3. Thermogravimetric Analysis (TGA)

Thermal degradation was performed using TGA via a TGA2 instrument (Mettler Toledo, Greifensee, Switzerland). The samples prepared from the cross-section (5 ± 0.5 mg) were heated from 50 °C to 600 °C under an N_2_ atmosphere at the heating ramp of 10 °C/min. The decomposition temperature was determined at 5% weight loss (T_5%_) and 50% weight loss (T_50%_). 

#### 2.3.4. Scanning Electron Microscopy (SEM) Images

The surface changes of PLA films were examined by field emission scanning electron microscopy (FE-SEM) using the Carl Zeiss ULTRA (Carl Zeiss, Oberkochen, Germany) microscope under an accelerated voltage of 5 kV. The trimmed samples was directly after conditioning (25°/96 h) fixed with carbon adhesive tape on aluminium holder and coated with 1 nm of platinum using Q150R ES (Quorum Technologies Ltd., Lewes, UK).

## 3. Results and Discussion

### 3.1. Determination of Ultimate Aerobic Biodegradability under Controlled Composting

The tested PLA films and reference microcrystalline cellulose (MCC) powder were exposed to controlled composting conditions. The evaluated biodegradation curves are shown in Figure 1. The MCC started intensively degrading after three days of the induction period (lag phase). The biodegradation degree was 72% after 45 days, which confirms the proper microbial activity of the used compost and the validity of the results (ISO 14855-1). Furthermore, an increased biodegradation rate following the addition of 10 wt.% ATBC plasticiser was observed when compared to neat PLA. The induction period of plasticised PLA (PLA/ATBC) was eight days shorter than for neat PLA films. The level of biodegradation after 30 days was 6% for neat PLA and 16% for PLA/ATBC. While the biodegradation of PLA/ATBC achieved a level of 97% (starting of the plateau phase) after 75 days, the biodegradation level of neat PLA was just 42% and according to the curse of biodegradation had not yet been finished. The low mineralisation rate (40 days lag phase, 90% biodegradation level within 120 days) of neat PLA exposed to aerobic thermophilic biodegradation was also reported by Itävaara et al. [20]. In the first 30 days, biodegradation results similar to those for plasticised PLA were obtained for ternary biocomposites based on PLA/ATBC/L-CNC. Since then, a slightly lower biodegradation rate was detected. The achieved biodegradation level after 75 days was 90%. The reason behind this phenomenon could be the inhibition of microbial activity due to the lignin coating of CNC. Yang et al. [30] and Micales et al. [31] reported that lignin inhibits the degradation of cellulose. The greatest influence on the biodegradation process was observed in ternary biocomposites containing CaCO_3_ nucleation agents. The induction period was 19 days shorter than for neat PLA films. A 58% level of biodegradation was achieved in 30 days, with 100% biodegradation and a plateau distribution recorded by the end of the experiment. This supports the findings of Hedge et al. [32], who observed increased biodegradation rates during anaerobic conditions after applying CaCO_3_ at low concentrations. The reason behind this could be that releasing CaCO_3_ to the biodegradation medium offers buffering action and helps to prevent the acidification of the pH [23] that could result from the degradation of a polymer matrix into lactic and succinic acids [32]. Regarding this point, the pH of individual composts was evaluated at the end of the experiment (see Table 3). In contrast to the findings for the blank vessels, an increase in the alkaline level of the compost was observed due to the introduction of PLA-based films. Greater increases were observed for plasticised PLA (PLA/ATBC) and ternary biocomposites (PLA/ATBC/CaCO_3_, PLA/ATBC/L-CNC). Another aspect that can influence the biodegradation rate is the surface treatment of CaCO_3_, as fatty acid-based sizing agents are commonly applied to prevent their agglomeration [33]. The fatty acid causes a chemical reaction with the ester linkage of PLA, which evokes chain scission and induces thermal degradation [34].

### 3.2. Analysis of Biodegradability in Freshwater Biotope

#### 3.2.1. Fourier Transform Infrared Spectroscopy (FT-IR)

The ATR-FT-IR spectra of PLA-based films (Figure 2a) showed a typical absorption band corresponding to the C=O stretching of ester groups at 1747 cm^−1^, asymmetric and symmetric CH_3_ stretching at 2995 cm^−1^ and 2945 cm^−1^. Furthermore, the C–O stretching bands of CH–O at 1180 cm^−1^ and O–C=O groups at 1127 cm^−1^, 1080 cm^−1^ and 1043 cm^−1^, respectively, were observed. The same absorption band characteristic was detected by Sessini et al. [35], Lee et al. [36], Weng et al. [37] and Amorin et al. [38]. Moreover, bending frequencies for CH_3_ have bends identified at 1452 cm^−1^, 1382 cm^−1^ and 1359 cm^−1^, as well as bends related to the C=O double-bound around 700 cm^−1^ [39]. Any significant differences were not detected in the absorption bands of plasticised PLA (PLA/ATBC) and PLA/ATBC/L-CNC ternary biocomposites. However, an increased peak height ratio of PLA/ATBC/CaCO_3_ ternary biocomposites around 1450 cm^−1^ and 872 cm^−1^ could be seen, as well as a decreased peak height ratio around 1750 cm^−1^. From the absorption band of PLA films exposed to 3 months of degradation in freshwater, it is evident that FTIR peak height ratios significantly decreased. Consequently, it can be stated that chemical changes were evoked by the degradation process. Moreover, a new peak at around 3400 cm^−1^ (Figure 2b) indicates the occurrence of hydrolysis degradation [40]. When comparing the neat PLA film spectra with PLA biocomposites based on ATBC, CaCO_3_ and L-CNC, the smaller decrease in peak intensity is evident. The lowest peak height ratios and the greatest chemical changes were observed in the PLA biocomposite containing the ATBC plasticiser and CaCO_3_ nucleating agents. To assess differences in degradation kinetics, the peak intensities at 1080 cm^−1^, 1127 cm^−1^, 1359 cm^−1^, 1382 cm^−1^, 1452 cm^−1^, 1747 cm^−1^ and 3400 cm^−1^ are compared in Table 4. Considerable differences in the kinetics of degradation can be seen from the evaluated peaks. The neat PLA showed a significant decrease in FTIR peak height ratios after only 3 months of degradation. However, an intensive decrease in peak height was observed for plasticised PLA after 2 months of exposition. The PLA/ATBC/L-CNC biopolymer showed increased resistance to degradation in the first month. Since then, an intensive decrease in peak height ratios was also observed. The greatest decrease of absorption bands was observed in ternary biocomposites based on PLA/ATBC/CaCO_3_ after 2 months of degradation.

#### 3.2.2. Differential Scanning Calorimetry (DSC)

The results for non-isothermal conditions during the first heating step were evaluated for processed samples (initial state) and samples degraded in a fresh water biotope over 3 months. The effect of processing parameters and the promoting effect of ATBC and/or nucleated agents (L-CNC, CaCO_3_) on the thermal parameters and crystallisation behaviour were investigated. As can be seen from the DSC curves in Figure 3, neat PLA, plasticised PLA (PLA/ATBC) and related ternary biocomposites (PLA/ATBC/CaCO_3_, PLA/ATBC/L-CNC) exhibited characteristic glass transition (~37–64 °C), cold crystallisation (~79–116 °C) and melting (~139–171 °C). A summary of the transition temperatures and related enthalpies is given in Table 5. The shift of the glass transition temperature (T_g_) towards a lower temperature (46.7 °C) compared to neat PLA (62 °C) could be observed after introducing 10 wt.% of ATBC to processed samples. This shift was caused by the decrease in intramolecular binding forces due to the addition of low molecular weight plasticiser, which increased the intermolecular distance and also resulted in an increase in macromolecular chain mobility [3]. Consequently, cold crystallisation temperature (T_cc_) and primary melting temperature (T_pm_) also reached lower values compared to neat PLA. Furthermore, dual melting peaks could be observed after the introduction of ATBC. Wu et al. [41] observed that broad cold crystallisation indicates the formation of multi-population lamella with different layer thicknesses that cause the dual melting peaks. These dual melting peaks can be ascribed to the formation of different crystalline structures. The thermal parameters during the non-isothermal processing conditions of PLA allow for the formation of two polymorphs: α and α′-crystallites. The non-isothermal crystallisation from relaxed melt at a temperature above 120 °C led to the formation of α-crystallites. Between 100 °C and 120 °C, both forms crystallise in coexistence; below 100 °C, only metastable α′-crystallites grow [42]. The packing density of α′-crystallites is lower due to conformational disorder. During the subsequent heating, α′-crystallites transformed into more stable α-crystallites at a temperature of around 150 °C [43]. The introduction of 1 wt.% L-CNC (PLA/ATBC/L-CNC) caused a slight increase in T_g_ and T_cc_ when compared to the plasticised system (PLA/ATBC). In our previous study [44], we reported the synergistic effect of 1 wt.% L-CNC and 10 wt.% ATBC on the crystallisation of the PLLA matrix during injection moulding. The nucleation efficiency of L-CNC on the PLA matrix has also been reported by Gupta et al. [15]. However, our results did not correlate with these previous reports. The fast cooling during the processing of ternary PLA/ATBC/L-CNC films—as well as different PLA matrices that contain more D-lactic acid than those used in previous reports—could be ascribed to this phenomenon. The addition of 10 wt.% of CaCO_3_ (PLA/ATBC/CaCO_3_) caused a further decrease in T_g_ (42.4°C) when compared to plasticised PLA (46.7 °C). The cold crystallisation temperature remained unchanged when compared to plasticised PLA. In contrast to the research of Suksut et al. [45], this indicates that CaCO_3_ does not function as a proper nucleating agent in the fast cooling conditions of the PLA matrix.

Furthermore, an increase in T_g_ during the three months of degradation for neat PLA could be observed. This indicates the crystallinity degree increased within this time. In contrast, plasticised PLA presented a decrease in T_g_ within the first two months. This could be ascribed to the chain shortening (hydrolysis reaction) of macromolecular chains within the amorphous region. An increase in T_g_ after three months of degradation within the freshwater biotope was observed. This phenomenon could be related to the degradation of the less accessible semicrystalline region over time and the releasing of the ATBC plasticiser and oligomeric PLA in the environment. A correlation in the shift of T_cc_ towards lower temperatures could also be observed. No significant changes within thermal transitions were observed during the degradation time of ternary PLA/ATBC/L-CNC films. This may be because of the inhibition of microbial activity due to the lignin coating of CNC as well as the steric hindrance of plasticised PLA. During the degradation of PLA/ATBC/CaCO_3_ films, T_g_ again decreased within the first two months and increased after the third month. This also correlates with the changes in T_cc_. Thus, the same mechanism of degradation is believed to exist as in plasticised PLA, the leaching of oligomers and ATBC in the environment. This has been further supported by CaCO_3_ and its fatty acids.

The crystallinity degree (X_C_) calculations shown in Table 5 can be considered authoritative only in the initial state after processing and, at most, during the first month of degradation. The later measured samples were degraded to the extent that we cannot confirm the exact weight content of additives. Consequently, the results for the second and third months were not described.

#### 3.2.3. Thermogravimetric Analysis (TGA)

The results of TGA are shown in Figure 4 and summarised in Table 6. A significant decrease in the thermal stability (initial decomposition temperature T_5_%) of PLA films was observed with the addition of the ATBC plasticiser. Similar results were observed by Maiza et al. [3] and Sessini et al. [35]. Nelson et al. [46] reported that a higher degree of crystallinity is directly involved in the higher thermal stability of L-CNCs. Our previous study [47] found that the presence of L-CNC in PLA increased the crystallinity degree, resulting in increased thermal stability. As stated in the DSC section, the low concentration (1 wt.%) of L-CNC did not increase the crystallinity degree in the chosen processing conditions. Despite this, an increase in the thermal stability of the PLA/ATBC/L-CNC biocomposites was observed. This could be due to the steric hindrance of plasticised PLA macromolecular chains by the high volume of L-CNC particles and their huge surface area. Nekhamanurak et al. [34] reported the profound impact of the fatty acid treatment of CaCO_3_ on the thermal stability of PLA/CaCO_3_ biocomposites. The chemical reaction between the fatty acid and ester linkage of PLA caused chain scission and induced thermal degradation. This is supported by the obtained results. The presence of 10 wt.% of CaCO_3_ in plasticised PLA caused the greatest decrease in thermal stability. The initial degradation temperature (T_5_%) decreased around 61 °C and temperature at the midpoint (T_50_%) around 41 °C. When the TGA results of PLA films exposed to freshwater biodegradation are compared, similar conclusions about the influence of additives can be made as for the thermophilic composting process. The minuscule decrease in the thermal degradation temperatures of neat PLA is a sign of biodegradation resistance. This supports the results of Itävaara et al. [20], who observed very slow PLA biodegradation in an aquatic test at room temperature. Kallel et al. [28] reported that the addition of plasticiser evokes faster hydrolysis and hydrolytic attack by a microorganism. Consequently, a significant decrease in the initial thermal degradation temperature was observed for PLA/ATBC. Moreover, the PLA/ATBC/CaCO_3_ biocomposites showed a significant decrease in initial decomposition temperature. This was due to intensive hydrolysis and surface erosion evident even after one month of exposition to freshwater biodegradation.

#### 3.2.4. Scanning Electron Microscopy (SEM) Images

The SEM images of the surfaces of PLA films after processing and by the end of the freshwater biodegradation process (after three months) are shown in Figure 5. The produced PLA films showed smooth surfaces. Significant differences in degradation were observed between neat PLA films and PLA films containing ATBC plasticisers and/or nucleation agents (L-CNC, CaCO_3_). Neat PLA films showed a relatively smooth surface with a small number of cavities. In contrast, PLA films with plasticiser showed a rough surface with many eroded pinholes. The PLA polymers had the characteristics of bulk (homogenous) degradation [18] and the continuous decreasing of weight and thickness without releasing carboxyl acid and hydroxyl by-products [48] until the critical sample thickness was achieved [49]. The increased molecular mobility following the addition of the ATBC plasticiser caused faster hydrolysis, which increased the rate of hydrolytic attack, surface erosion and microbial enzymatic degradation. Kallel et al. [28] reported increased surface erosion as a result of faster hydrolysis and higher microbial enzyme activity following the exposition of a PLA/PEG blend to an aerobic liquid medium.

As previously stated, the considerable effect of the lignin coating of CNC in PLA/ATBC/L-CNC improved thermal stability and inhibited degradation in the first month of freshwater biotope conditions. However, when the SEM images of the PLA/ATBC biopolymers and PLA/ATBC/L-CNC surfaces by the end of the biodegradation experiment (after three months) are compared, no significant differences can be observed. The largest pinholes and highest surface roughness were observed in the PLA/ATBC/CaCO_3_ material combination. The ATBC plasticiser and fatty acid treatment of CaCO_3_ resulted in intensive hydrolysis that increased surface erosion and caused the releasing of CaCO_3_. The release of the CaCO_3_ additive led to a further increase in surface roughness, which ensured faster hydrolytic and enzymatic degradation of this biocomposite.

## 4. Conclusions

The influence of additives such as natural-based plasticiser ATBC, CaCO_3_ and L-CNC on the biodegradation of PLA biocomposite films in thermophilic compost and freshwater biotope has been studied. Very low biodegradation rates were observed for neat PLA. The application of ATBC at a 10 wt.% loading significantly increased the induction period and the level of biodegradation of PLA films in controlled composting (ISO 14855-1). Slightly lower biodegradation rates were detected in ternary biocomposites containing L-CNC (1 wt.%). The highest biodegradation rates were observed in PLA/ATBC/CaCO_3_ biocomposites. The induction period was 19 days shorter than for neat PLA films, and 94% biodegradation in 60 days was observed. In contrast, neat PLA exposed to the same conditions achieved only 16% biodegradation. Furthermore, the results of qualitative analyses (SEM, DSC, TGA and FTIR) of the biodegradation of PLA films in freshwater biotope showed that the biodegradation rate significantly increased with the addition of ATBC to the PLA matrix. Considerable changes in surface roughness, chemical composition and thermal properties were detected after three months of biodegradation. Moreover, the inhibiting effect of the lignin coating of CNC on the biodegradation of PLA ternary biocomposites was observed in the first months. Also in freshwater biotope, the highest biodegradation rate has been detected in PLA/ATBC/CaCO_3_ biocomposites. Significant changes in chemical composition and thermal properties were observed even after one month of biodegradation.

This study found that the biodegradation potential of the PLA/ATBC/CaCO_3_ biocomposites significantly increased due to the synergistic influence of both additives. In the chosen processing conditions, the ATBC plasticiser had a more profound effect on the enhancement of the crystallinity degree than both heterogeneous nucleation agents (CaCO_3_ and L-CNC). Furthermore, the thermal stability of the ternary biocomposites based on PLA/ATBC/CaCO_3_ fell by 60 °C and 27 °C when compared to neat PLA and plasticised PLA/ATBC, respectively.

## Figures and Tables

**Figure 1 polymers-13-00594-f001:**
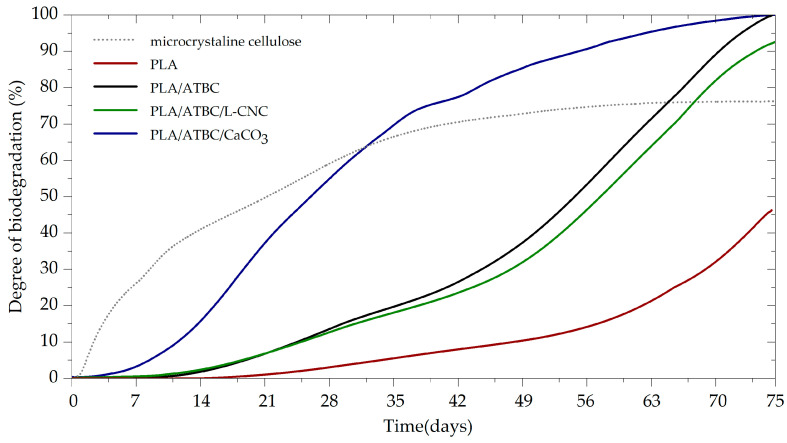
Biodegradation curves of PLA biocomposites under controlled composting ISO 18455-1.

**Figure 2 polymers-13-00594-f002:**
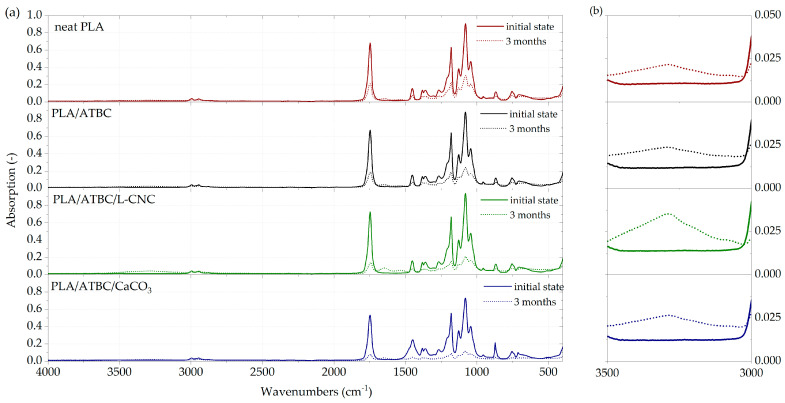
FTIR spectra of PLA biocomposites. (**a**) At initial state and after three months of biodegradation in freshwater biotope. (**b**) Occurrence of hydrolysis degradation.

**Figure 3 polymers-13-00594-f003:**
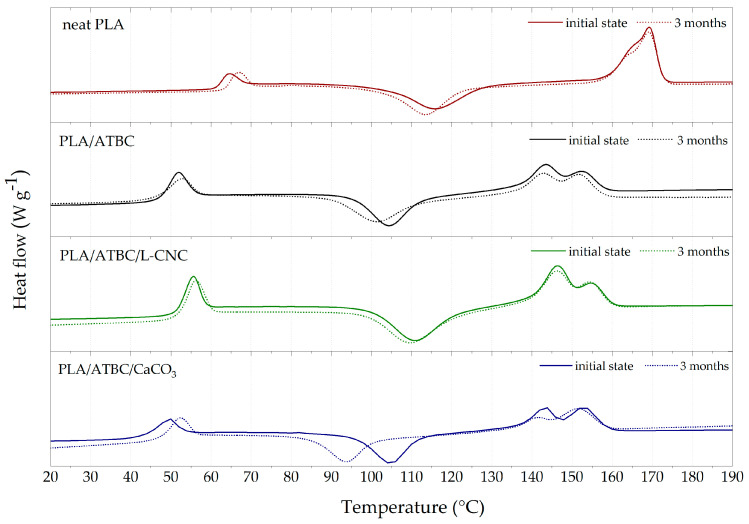
Thermal analysis (DSC) curves of PLA biocomposites at initial state and after three months of biodegradation in freshwater biotope.

**Figure 4 polymers-13-00594-f004:**
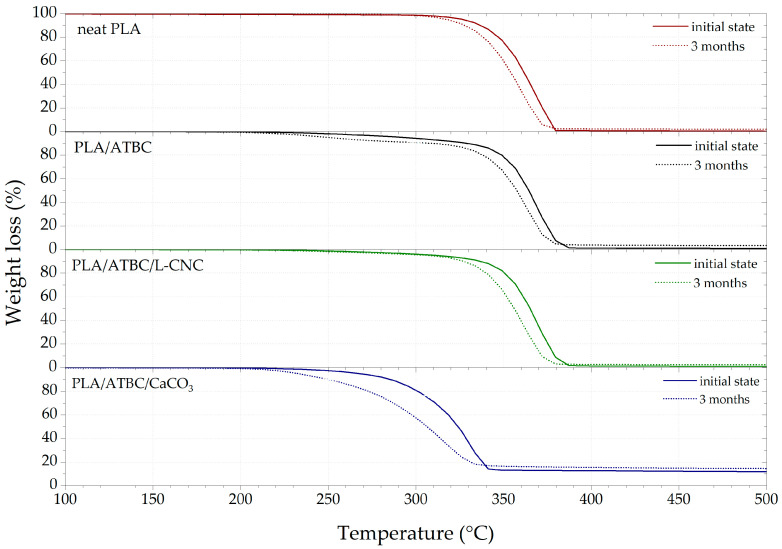
Thermogravimetric analysis (TGA) curves of PLA biocomposites at initial state and after 3 months of biodegradation in freshwater biotope.

**Figure 5 polymers-13-00594-f005:**
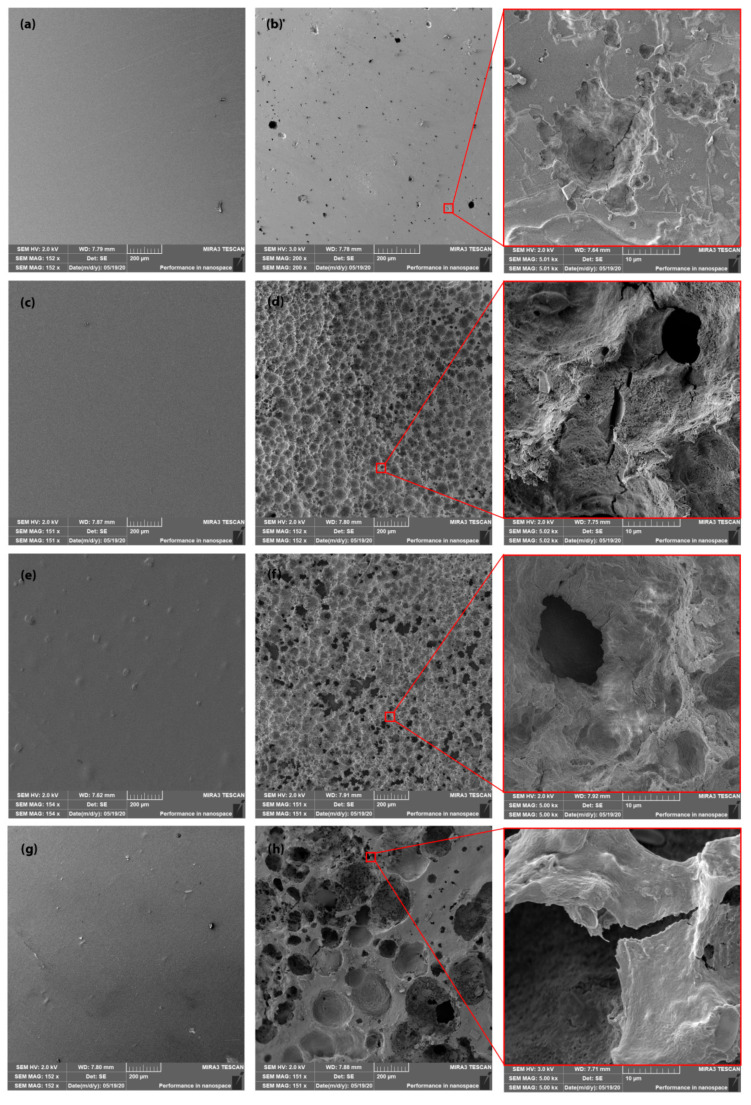
SEM images of (**a**) neat PLA film at initial state, (**b**) neat PLA film after three months of biodegradation, (**c**) PLA/ATBC at initial state, (**d**) PLA/ATBC after three months of biodegradation, (**e**) PLA/ATBC/L-CNC at initial state, (**f**) PLA/ATBC/L-CNC after three months of biodegradation, (**g**) PLA/ATBC/CaCO_3_ at initial state and (**h**) PLA/ATBC/CaCO_3_ after three months of biodegradation.

**Table 1 polymers-13-00594-t001:** Sample compositions.

Sample Designation	Composition (wt.%)			
	PLA	ATBC	L-CNC	CaCO_3_
PLA	100	-	-	-
PLA/ATBC	90	10	-	-
PLA/ATBC/L-CNC	89	10	1	-
PLA/ATBC/CaCO_3_	80	10	-	10

**Table 2 polymers-13-00594-t002:** The individual proportions of total organic carbon of PLA biocomposite components.

**Sample Designation**	Proportions (%)
PLA	50.0
ATBC	59.7
L-CNC	44.4
CaCO_3_	-

**Table 3 polymers-13-00594-t003:** Evaluated pH of compost at the end of the experiment.

Sample Designation	pH (-)
Blank vessel	6.5
PLA	7.0
PLA/ATBC	7.3
PLA/ATBC/L-CNC	7.2
PLA/ATBC/CaCO_3_	7.5

**Table 4 polymers-13-00594-t004:** Peak intensity of PLA biocomposites exposed to 3 months of biodegradation in freshwater biotope.

Sample Designation	Exposition Time (Months)	3400 cm^−1^	1747 cm^−1^	1452 cm^−1^	1382 cm^−1^	1359 cm^−1^	1127 cm^−1^	1080 cm^−1^
PLA	Initial state	0.01	0.68	0.15	0.14	0.14	0.38	0.91
	1	0.02	0.53	0.15	0.13	0.13	0.34	0.73
	2	0.02	0.38	0.11	0.10	0.10	0.27	0.56
	3	0.02	0.21	0.07	0.07	0.7	0.16	0.31
PLA/ATBC	Initial state	0.01	0.68	0.15	0.14	0.14	0.39	0.89
	1	0.02	0.57	0.15	0.13	0.13	0.36	0.77
	2	0.02	0.22	0.08	0.08	0.07	0.16	0.29
	3	0.02	0.19	0.07	0.07	0.06	0.13	0.29
PLA/ATBC/L-CNC	Initial state	0.01	0.72	0.15	0.14	0.15	0.4	0.93
	1	0.02	0.64	0.15	0.14	0.14	0.38	0.86
	2	0.02	0.22	0.08	0.07	0.07	0.15	0.28
	3	0.03	0.14	0.07	0.08	0.06	0.12	0.22
PLA/ATBC/CaCO_3_	Initial state	0.01	0.54	0.24	0.15	0.14	0.35	0.73
	1	0.02	044	0.21	0.13	0.13	0.31	0.63
	2	0.02	0.15	0.07	0.06	0.06	0.12	0.2
	3	0.02	0.08	0.05	0.04	0.04	0.07	0.12

**Table 5 polymers-13-00594-t005:** Thermal analysis (DSC) data of PLA biocomposites exposed to 3 months of biodegradation in freshwater biotope.

Sample Designation	Exposition Time (Months)	T_g_ (°C)	T_cc_ (°C)	∆H_cc_ (J/g)	T_pm_ (°C)		∆H_m_ (J/g)	∆H_mα’_ (J/g)	X_C_ (%)
PLA	Initial state	62.0	115.7	35.62	169.2		36.3	-	0.8
	1	63.1	115.2	33.98	170.6		37.7	-	4.0
	2	63.4	114.2	32.20	170.1		36.6	-	-
	3	64.0	113.5	31.02	169.4		36.6	-	-
PLA/ATBC	Initial state	46.7	104.5	22.64	143.7	152.5	26.1	16.4	4.2
	1	37.6	87.3	20.63	139.7	151.2	25.9	9.4	6.3
	2	38.8	89.3	20.94	139.1	149.7	26.4	10.0	-
	3	45.1	101.5	22.78	142.8	151.8	26.0	14.6	-
PLA/ATBC/L-CNC	Initial state	51.2	110.8	25.70	146.5	154.9	25.8	17.7	0.1
	1	52.5	110.5	26.01	147.2	155.1	25.8	18.2	0.0
	2	52.8	110.8	26.77	147.7	155.3	26.3	18.2	-
	3	51.1	109.3	26.52	146.4	154.6	25.8	16.9	-
PLA/ATBC/CaCO_3_	Initial state	42.4	104.7	22.51	143.4	153.0	24.7	12.9	2.9
	1	38.4	81.5	15.81	137.4	149.0	26.0	6.8	13.7
	2	37.4	78.5	15.96	140.3	150.1	22.9	7.9	-
	3	46.7	93.8	19.15	141.2	151.4	22.4	9.1	-

**Table 6 polymers-13-00594-t006:** Thermogravimetric analysis (TGA) data of PLA biocomposites exposed to three months of biodegradation in freshwater biotope.

Sample Designation	Exposition Time (Months)							
	Initial State		1		2		3	
	T_5_ (%)	T_50_ (%)	T_5_ (%)	T_50_ (%)	T_5_ (%)	T_50_ (%)	T_5_ (%)	T_50_ (%)
PLA	329.2	361.9	326.8	360.5	326.7	360.0	319.1	353.9
PLA/ATBC	294.3	364.3	282.2	359.9	274.5	357.8	250.9	356.8
PLA/ATBC/L-CNC	312.9	364.9	312.4	361.2	309.7	359.1	308.1	355.6
PLA/ATBC/CaCO_3_	267.8	320.7	247.1	299.9	233.8	299.5	233.5	299.4

## Data Availability

The data presented in this study are available on request from the corresponding author.

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
