# Peer review of "Biodegradation of Poly(Lactic Acid) Biocomposites under Controlled Composting Conditions and Freshwater Biotope"

_polymers, 2021, doi:10.3390/polym13040594_

Round 1
Reviewer 1 Report
This manuscript reports the effect of ATBC, L-CNC and CaCO3 on the biodegradation of PLA by using aerobic thermophilic composting and biotope conditions, respectively. The addition of ATBC and CaCO3 show higher degradation rate from both degradation conditions by using SEM, DSC, TGA and FTIR characterization. Overall, this manuscript is in high quality and the following comments should be considered before acceptance. 1. The x axis of Figure 1 should be modified to use either counted days or real date. 2. It is better not to show the crystallinity data of 2 and 3 months for PLA composites as the weight content of additives can not be confirmed.Author Response
Point 1: The x axis of Figure 1 should be modified to use either counted days or real date.
Response 1: I agree. I corrected Figure 1.
Point 2: It is better not to show the crystallinity data of 2 and 3 months for PLA composites as the weight content of additives can not be confirmed.
Response 2: I agree. I did not describe the crystallinity data of the second and third months and corrected the explanation (Track Changes - violet colour). Please see part crystallinity degree (Xc) calculations (chapter 3.3.2 Differential Scanning Calorimetry (DSC)).
Point 3: English language and style are fine/minor spell check required
Response 3: We send the manuscript to moderate revision of English structure and grammar.

Reviewer 2 Report
The manuscript describes the preparation of biopolymer membranes of PLA added with plastifiers and calcium ions to improve the mechanical properties and biodegradability of the composites. The authors use two approaches to test biological degradability: respirometry and the polymers' evaluation after three months of incubation in an aquatic-model system.
For the evaluation with commercial compost, respirometry is used to assess the degradation kinetics, showing the results in a figure, where differences are observed among the different membranes tested. It would be interesting to have a measure of variability (standard deviation), even if the experiment was done only in duplicate.
Several analytical methods are used for biodegradability in freshwater, including IR, DSC, and TGA. The methods are really useful to describe the membranes, their predominant organic groups, or thermal stability, but not on the compounds' biodegradation. Electronic microscopies demonstrated that the structure of the membranes was affected by biological degradation, even though the results of other tests were a slight modification on the properties tested. I am not saying that the results are not important and should not be reported, but I would expect other measurements to assess biodegradation.
A simple difference in weight can give an idea of the percentage of biodegradation of the membranes. Can the authors please comment on this? or if by any chance do they have the information, can you include it?
In material and methods, can you please expand the methods used for membrane preparation for SEM analysis. I would expect at least a few microbial cells to be observed, but I guess it depends on the method of sample preparation.
I recommend that identifying the different treatments and analysis is included in the description of figures and tables.
The document needs a moderate revision of English structure and grammar.
Author Response
Dear Reviewer,
Thank you for your review. We responded to your comments and upload it as a world file. Please see the attachment.
with best regards
Pavel Brdlík

Round 2
Reviewer 2 Report
The authors have addressed my suggestions. I consider that is not necessary to include the information provided in table X (included in the response document) as supplemental material.
I look forward to the document with identification of microorganisms present in the membranes.